

# Climate, soil or both? Which variables are better predictors of the distributions of Australian shrub species?

Yasmin Hageer, Manuel Esperón-Rodríguez, John B. Baumgartner and Linda J. Beaumont

Department of Biological Sciences, Macquarie University, Sydney, New South Wales, Australia

## ABSTRACT

**Background**. Shrubs play a key role in biogeochemical cycles, prevent soil and water erosion, provide forage for livestock, and are a source of food, wood and non-wood products. However, despite their ecological and societal importance, the influence of different environmental variables on shrub distributions remains unclear. We evaluated the influence of climate and soil characteristics, and whether including soil variables improved the performance of a species distribution model (SDM), Maxent.

**Methods**. This study assessed variation in predictions of environmental suitability for 29 Australian shrub species (representing dominant members of six shrubland classes) due to the use of alternative sets of predictor variables. Models were calibrated with (1) climate variables only, (2) climate and soil variables, and (3) soil variables only.

**Results**. The predictive power of SDMs differed substantially across species, but generally models calibrated with both climate and soil data performed better than those calibrated only with climate variables. Models calibrated solely with soil variables were the least accurate. We found regional differences in potential shrub species richness across Australia due to the use of different sets of variables.

**Conclusions**. Our study provides evidence that predicted patterns of species richness may be sensitive to the choice of predictor set when multiple, plausible alternatives exist, and demonstrates the importance of considering soil properties when modeling availability of habitat for plants.

## INTRODUCTION

Species distribution models (SDMs) are tools used to assess the spatial distribution of potentially suitable habitat for species, and to hypothesise how suitability is affected by environmental change (*Guisan & Thuiller, 2005*). These tools generally correlate species' occurrence patterns with environmental variables, which are frequently selected from a set of 19 'bioclimatic' indices (*Nix, 1986*) available in WorldClim (*Hijmans et al., 2005*).

Although climate is recognized as a major factor controlling species' distributions (*Brown & Gibson, 1983*; *Woodward, 1987*), climate variables are unlikely to be the only relevant predictors of habitat availability (*Chatfield et al., 2010*; *Austin & Van Niel, 2011*), as plant survival and reproduction also depends on light, nutrients, water, and $CO_2$,

Corresponding author
Yasmin Hageer,
yasmin.hageer@gmail.com

as well as disturbances and biotic interactions (*Hibbard et al., 2001*; *Neher et al., 2004*; *Jackson, 2009*).

However, comparatively a few SDM studies directly incorporate non-climatic environmental variables such as soil properties (but see *Fitzpatrick et al., 2008*; *Martinson et al., 2011*; *Zhou et al., 2012*; *Condit et al., 2013*; *Taylor & Kumar, 2013*), even though the soil properties are known to considerably impacts the distribution of plant species (*Elmendorf & Moore, 2008*; *Dubuis et al., 2013*) for its importance to the plants as source of water and nutrients (*Aerts & Chapin, 2000*) and physically for supporting root growth (*Martre et al., 2002*). Furthermore, some studies incorporated other factors in SDMs such as irradiance (see *Franklin, 1998*; *Summers et al., 2012*) as a light source for the plants, topography (*Franklin, 1998*; *Hosseini et al., 2013*), and landuse (*Meier et al., 2012*; *Stanton et al., 2012*; *Titeux et al., 2016*) in which degradation in plant habitats and loss of plant biodiversity is strongly influenced by changes in landuse and increase of urbanization is considered (*Lawler et al., 2014*). This poorly integration of non-climatic factors in modelling studies may partly reflect difficulties with obtaining appropriate data sets at relevant spatial scales, particularly with regards to soil variables that are related to plant functionality. It is highly recommended by all these modelling studies that SDMs should be calibrated with physiologically-relevant environmental variables, as this should lead to SDMs with greater predictive power (*Austin, 2002*; *Austin, 2007*; *Williams et al., 2012*).

Whilst not a strict botanical category, shrubs are generally regarded as low height, woody perennial plants with several base-stems (*Zeng, Zeng & Barlage, 2008*; *Meng, Ni & Harrison, 2009*). As the dominant flora in arid and semi-arid regions, shrubs play a key role in enhancing soil fertility, reducing runoff, soil loss (*Pressland, 1973*; *Xu et al., 2008*; *Song et al., 2013*) and dust emissions (*Engelstaedter et al., 2003*), and sequestering carbon in grassland ecosystems (*Yashiro et al., 2010*). By providing fodder for livestock (*Lefroy et al., 1992*), shrubs can enhance economic returns for dryland farms by providing an 'out-of-season' food source (*Monjardino, Revell & Pannell, 2010*).

The distribution of shrub species is strongly influenced by environmental conditions, such as climate (*Pedley, 1979*; *Westman, 1991*; *Kienast, Wildi & Brzeziecki, 1998*). Plant species occurring in arid to semi-arid regions have evolved several traits enabling them to tolerate extended periods of low precipitation and high temperature. These include small leaves (*Smith, Monson & Anderson, 1997*), slower growth rates, and more horizontal, rather than vertical, growth (*Zeng, Zeng & Barlage, 2008*). During the hot and dry season, stomata may be partly closed, reducing transpiration and water loss, leaves may be shed (*Smith, Monson & Anderson, 1997*), and physiological activity is restricted (*Reynolds, 1999*). Following rainfall events, leaves expand and stomata fully open (*Zeng, Zeng & Barlage, 2008*), and the negative impacts of the dry season may be compensated for via enhanced physiology and growth (*Reynolds, 1999*). Shrubs also have a deeper, wider rooting system than grasses, enabling the efficient extraction of water in low moisture environments (*Burgess, 1995*).

Physical and chemical soil properties, and biotic interactions play a major role in controlling the distribution of shrub species (*Pedley, 1979*). Shrubs usually occur on shallow, coarse and infertile soils (*Groves, 1994*), and are adapted to live on sandy soils with
limited moisture. Shrubs often accumulate their organic matter beneath their canopies, thereby enriching the nutrient pool horizontally, enabling these species to grow in infertile soils (*Zinke, 1962*; *Jackson & Caldwell, 1993*; *Schlesinger et al., 1996*; *Burke et al., 1998*) and providing microclimatic conditions that stimulate microbial biomass and activity (*Sandoval Pérez et al., 2016*).

In this study, we assessed the extent to which soil variables, in conjunction with climate, may increase the predictive power of habitat suitability models of Australian shrub species. We hypothesised that models calibrated with both climate variables and soil properties will have greater predictive power compared with models that incorporate only climate or soil parameters. To test this hypothesis, we selected as a case study 29 shrub species that together span the distribution of major shrubland vegetation types across the continent.

## METHODS

### Species data

Shrubs are recognized as plants that are woody, "multi-stemmed at the base (or within 200 mm from ground level) or single stemmed, and less than 2 m" in height (*ESCAVI, 2003*, p. 87). Species are grouped into five growth forms: acacia, mallee (*Eucalypt* species), heath (which typically belong to Epacridaceae, Myrtaceae, Fabaceae and Proteaceae), chenopods, and samphire. Combined, these five shrub growth forms occupy a substantial part of the Australian landmass, mainly in semi-arid and arid regions, which form ca. 70% of the continent.

We identified 29 shrub species for inclusion in this study (Table 1), based on their dominance and endemism. Combined, these species represent the variety of shrub growth forms present on the continent. We obtained occurrence records from the Atlas of Living Australia (ALA, see http://www.ala.org.au/). Prior to downloading records, we applied filters to exclude records that did not contain coordinates (an average of 2% of records per species), were collected before 1960, or were identified by ALA as environmental outliers given the climatic envelope of the species. This resulted in an average of 3,523 (SD = 3,214) records per species.

### Selection of predictor variables

We considered annual, seasonal and monthly climate variables known to influence the distribution of shrubs (e.g., *Xin-Rong, 2001*; *Li et al., 2009*; *Gherardi & Sala, 2015*). Gridded data for nineteen climate variables, developed by the Wallace Initiative (http://wallaceinitiative.org), were downloaded at a resolution of $0.05 \times 0.05$ arc-minutes. These data were derived from spatially interpolated monthly precipitation and temperature observations (baseline period 1976–2005) obtained from the Australian Water Availability Project (AWAP, *Raupach et al., 2009*; *Raupach et al., 2012*; http://www.bom.gov.au/jsp/awap/) (for more details see *Vanderwal et al., 2011*). Multicollinearity of variables can result in over-fitting of SDMs and complicate interpretation of variables' contributions (*Elith, Kearney & Phillips, 2010*; *Williams et al., 2012*; *Zhou et al., 2012*), therefore we assessed pair-wise correlations among variables. When Pearson's correlation coefficients were greater than 0.85 we removed one of the variables. This reduced the number of climate

**Table 1  Distribution changes in shrubs suitable habitats using the models of climate only variables vs. climate with-soil-variables.** The projected area (km²) of suitable habitat for 29 Australian shrub species, based on models using climate-only variables ($V_C$), and using climate-with-soil variables ($V_{C+S}$). Also shown is the percentage of $V_C$ habitat that is also suitable according to $V_{C+S}$ models (Overlap), the percentage of $V_{C+S}$ habitat that is not suitable in the corresponding $V_C$ model (Gain), and the percentage of $V_C$ habitat that is not suitable in the corresponding $V_{C+S}$ model (Loss).

| Family | Scientific authority | Species | $V_C$ (km²) | $V_{C+S}$ (km²) | Overlap (%) | Gain (%) | Loss (%) |
|---|---|---|---|---|---|---|---|
| Asteraceae | DC | *Ozothamnus turbinatus* | 170,433 | 135,372 | 73.1 | 7.9 | 26.9 |
| Casuarinaceae | (Diels) LAS Johnson | *Allocasuarina campestris* | 853,151 | 479,167 | 50.5 | 10.2 | 49.5 |
| Chenopodiaceae | Benth. | *Atriplex angulate* | 2,308,110 | 1,051,859 | 42.4 | 6.9 | 57.6 |
| | Aellen | *Atriplex eardleyae* | 1,800,784 | 710,007 | 27.8 | 29.6 | 72.2 |
| | F.Muell. | *Atriplex holocarpa* | 1,843,472 | 1,239,199 | 59.1 | 12.0 | 40.9 |
| | Lindl. | *Atriplex nummularia* | 2,895,215 | 1,633,454 | 52.0 | 7.8 | 48.0 |
| | Heward ex Benth. | *Atriplex vesicaria* | 1,969,767 | 1,762,939 | 82.4 | 7.9 | 17.6 |
| | (R.Br.) Paul G. Wilson | *Maireana aphylla* | 1,899,906 | 801,821 | 31.4 | 25.7 | 68.6 |
| | Labill. | *Epacris impressa* | 271,063 | 242,672 | 84.2 | 5.9 | 15.8 |
| | F.Muell. ex Benth. | *Acacia aneura* | 3,556,328 | 3,223,147 | 87.3 | 3.7 | 12.7 |
| | F.Muell. | *Acacia sclerosperma* | 1,346,499 | 1,535,144 | 89.9 | 21.1 | 10.1 |
| | F.Muell. | *Acacia tetragonophylla* | 3,124,692 | 2,161,370 | 66.4 | 4.1 | 33.6 |
| | Benth. | *Acacia victoriae* | 3,514,249 | 2,194,372 | 56.1 | 10.2 | 43.9 |
| | Bonpl. | *Eucalyptus diversifolia* | 231,681 | 180,003 | 74.8 | 3.8 | 25.2 |
| | A. Cunn. ex J. Oxley | *Eucalyptus dumosa* | 856,660 | 493,203 | 53.2 | 7.6 | 46.8 |
| | F.Muell. | *Eucalyptus gracilis* | 737,006 | 393,675 | 49.6 | 7.1 | 50.4 |
| Myrtaceae | Labill. | *Eucalyptus incrassata* | 476,528 | 225,939 | 44.0 | 7.3 | 56.0 |
| | F.Muell. ex Miq. | *Eucalyptus oleosa* | 847,148 | 544,794 | 59.8 | 7.0 | 40.2 |
| | F.Muell. ex Miq. | *Eucalyptus socialis* | 1,941,927 | 1,285,483 | 62.2 | 6.1 | 37.8 |
| | Joy Thomps. | *Leptospermum continentale* | 328,309 | 201,695 | 54.1 | 12.0 | 45.9 |
| | S.Schauer | *Leptospermum glaucescens* | 1,892,279 | 1,507,507 | 69.8 | 12.4 | 30.2 |
| | (Gaertn.) F.Muell. | *Leptospermum laevigatum* | 423,574 | 263,262 | 58.5 | 5.8 | 41.5 |
| | (Sol. ex Ait.) Sm. | *Leptospermum lanigerum* | 179,713 | 141,520 | 76.2 | 3.3 | 23.8 |
| | J.R.Forst. & G.Forst. | *Leptospermum scoparium* | 73,399 | 72,442 | 95.0 | 3.7 | 5.0 |
| | Sm. | *Melaleuca ericifolia* | 340,373 | 234,349 | 60.0 | 12.9 | 40.0 |
| | Labill. | *Melaleuca squamea* | 91,872 | 100,253 | 94.0 | 13.9 | 6.0 |
| | Donn. ex Sm. | *Melaleuca squarrosa* | 177,074 | 150,365 | 78.5 | 7.5 | 21.5 |
| Sapindaceae | (F.Muell.) F.Muell. ex Benth. | *Atalaya hemiglauca* | 5,463,020 | 3,736,940 | 65.6 | 4.0 | 34.4 |
| Scrophulariaceae | F.Muell. | *Eremophila freelingii* | 1,663,875 | 647,367 | 31.4 | 19.2 | 68.6 |

variables to a set of five: Mean Annual Temperature ($T$), Maximum Temperature of the Warmest Month (TM warm), Total Annual Precipitation (P), Precipitation of the Warmest Quarter (PQ warm), and Precipitation of the Coldest Quarter (PQ cold). See correlation matrix of the variables in Table S1 .

Spatial data describing four soil variables that capture soil functionality (*Sauer, Cambardella & Meek, 2006*; *Fisher, Whittaker & Malhi, 2011*; *Meier et al., 2012*) were obtained from the CSIRO Data Access Portal at a resolution of 3 × 3 arc-seconds (~90 × 90 m) (http://www.clw.csiro.au/aclep/soilandlandscapegrid/index.html). This data include: percent clay content (CLAY; *Viscarra Rossel et al., 2014a*); bulk density (BD; *Viscarra Rossel*

**Table 2  The environmental predictor sets used in the different models.** Alternative predictor sets used in models.

| Abbreviation | Environment variable | $V_C$ | $V_{C+S}$ | $V_S$ |
|---|---|:---:|:---:|:---:|
| $T$ | Annual mean temperature | ● | ● | |
| TM warm | Maximum temperature of warmest month | ● | ● | |
| P | Mean annual precipitation | ● | ● | |
| PQ warm | Precipitation of warmest quarter | ● | ● | |
| PQ cold | Precipitation of coldest quarter | ● | ● | |
| BD | Bulk soil density (g/cm³) | | ● | ● |
| CLAY | Clay content percentage | | ● | ● |
| pH | pH CaCl$_2$ | | ● | ● |
| OC | Organic carbon percentage | | ● | ● |

**Notes.**
$V_C$, climate variables only; $V_{C+S}$, climate and soil variables; $V_S$, soil variables only.

*et al., 2014b*), which reflects soil porosity; pH CaCl$_2$ (pH; *Viscarra Rossel et al., 2014c*), which reflects soil acidity; and organic carbon (OC; *Viscarra Rossel et al., 2014d*). These variables describe the corresponding physical and chemical soil characteristics that are known to influence vascular plant growth and distribution (e.g., *Jarvis, 1974*; *Crawley, 1997*), and do not correlate highly to each other (Table S1). For each soil variable used in this study, the three soil depth layers (0–5 cm; 5–15 cm; 15–30 cm) were highly correlated ($|r| > 0.98$), thus, we used measurements from the first layer only. Using ArcGIS v10.4 (ESRI Inc., 2010), all soil data were resampled to a resolution of $0.05 \times 0.05$ arc-minutes using bilinear interpolation, thereby matching the resolution of the climate data. Finally, each predictor variable was projected to a $5 \times 5$ km equal area grid (Australian Albers, EPSG: 3577).

## Generating Maxent models of shrub species' distributions

Three sets of models were calibrated for each species. Models referred to as $V_{C+S}$ were calibrated with climate and soil variables, $V_C$ models were calibrated with climate variables only, and $V_S$ models used only soil variables (Table 2).

We used Maxent (version 3.3.3k; *Phillips, Dudík & Schapire, 2004*; *Phillips, Anderson & Schapire, 2006*) to develop SDMs for all 29 species. Maxent is a presence-only modelling approach that produces a continuous probability field that can be interpreted as a relative index of environmental suitability. Higher values represent greater suitability of a region for the target species (*Phillips, Dudík & Schapire, 2004*; *Phillips, Anderson & Schapire, 2006*). A more detailed description of Maxent can be found in *Elith et al. (2011)*.

Models were initially fit using default settings. We then explored how different mathematical transformations of predictor variables ("features") influenced model predictions, and concluded that superior models were obtained when linear, quadratic, and product features were used. For each species, we generated a unique set of background points by identifying the subset of biogeographic subregion polygons (*IBRA, 2015* version 7.0) that contained occurrence points for the species, and randomly selecting up to 10,000 occurrence records from all plant records that fell within that subset of polygons (i.e., a spatially-constrained target-group background sample). This approach aims to balance environmental sampling biases between the modelled species and the background records

required by Maxent for model calibration (*Warren, Glor & Turelli, 2008*; *Phillips et al., 2009*; *Merow, Smith & Silander, 2013*).

To reduce bias caused by randomly selecting occurrence records for model training, we generated five cross-validated replicate models for each species, using a different subset of 20% of occurrence records to test each model. For each species, the five replicate models were projected to the model-fitting predictor grids, and these five projections were then averaged to produce the final projection.

## Model performance

Currently, there is no ideal approach for evaluating model performance, although the area under the receiver-operating characteristic curve (AUC) is the most common measure of the performance of Maxent (*Merow, Smith & Silander, 2013*). A high AUC score indicates that the model can distinguish between presence and background points, with model performance generally considered good when AUC >0.75 (*Pearce & Ferrier, 2000*; *Elith et al., 2006*). Here, we used AUC as an indicator of model performance when models were calibrated with different sets of environmental factors. For each species we calculated the average test AUC over cross-validation replicates for each candidate model ($V_C$, $V_S$, $V_{C+S}$). We then identified which of the models led to the highest AUC. To test whether increasing number of variables in the $V_{C+S}$ would affect the value of AUC compared to the other models ($V_C$, $V_S$), we computed the number of parameters used in each fitted model using lambdas file corresponds to the number of features in the model and correlated the number of parameters with the AUC score for each species in the model. In addition, for each model we calculated the maximum value of the True Skill Statistic (TSS) on test data. Unlike AUC, which is threshold-independent, TSS reflects a model's sensitivity and specificity at a particular threshold. As such, it was proposed as an appropriate measure of model performance when the modelling application requires dichotomous maps of habitat suitability (*Allouche, Tsoar & Kadmon, 2006*). TSS values range from −1 to 1, where 1 implies perfect sensitivity and specificity, and values of zero or less indicate considerable commission and/or omission errors.

## Statistical analyses

Maxent suitability scores were converted to binary suitability (suitable/unsuitable) using the maximum training sensitivity plus specificity threshold (as recommended by *Liu, White & Newell, 2013*), which is numerically equivalent to the threshold corresponding to the maximum TSS. We then calculated pair-wise differences in the total area of suitable habitat predicted by the three sets of Maxent models using the ArcGIS extension SDMtoolbox (*Brown, 2014*). We also calculated potential shrub species 'richness', i.e., the number of species for which a grid cell was classified as suitable. These are not maps of species richness per se; rather, they are estimates of how many of the 29 species a given grid cell may be suitable for.

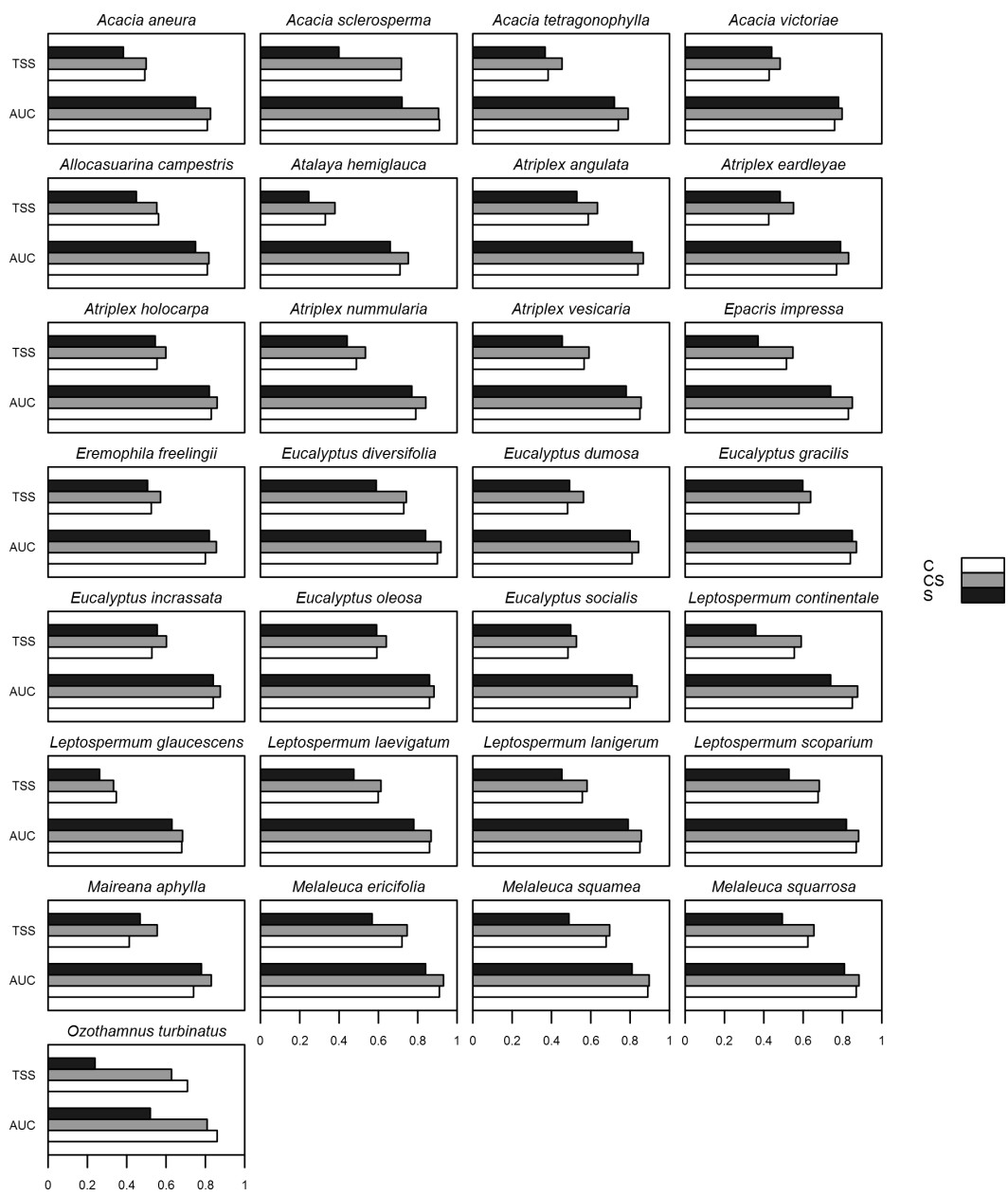

**Figure 1** **Mean values of the true skill statistic (TSS) and the area under the receiver-operating characteristic curve (AUC) of different used models.** Values of mean TSS and AUC for 29 shrub species from Australia for which suitable habitat was modelled (with Maxent) using three sets of predictor variables: climate-only ($V_C$), climate-with-soil ($V_{C+S}$), and soil-only ($V_S$).

# RESULTS

## Model performance

Of the 29 species, $V_{C+S}$ models had the highest mean AUC and TSS for 27 and 25 species, respectively (Fig. 1). Across all species, AUC was, on average, 0.848 (SD = 0.0139), 0.823 (0.0137), and 0.773 (0.0187) for $V_{C+S}$, $V_C$ and $V_S$ models, respectively, while the equivalent

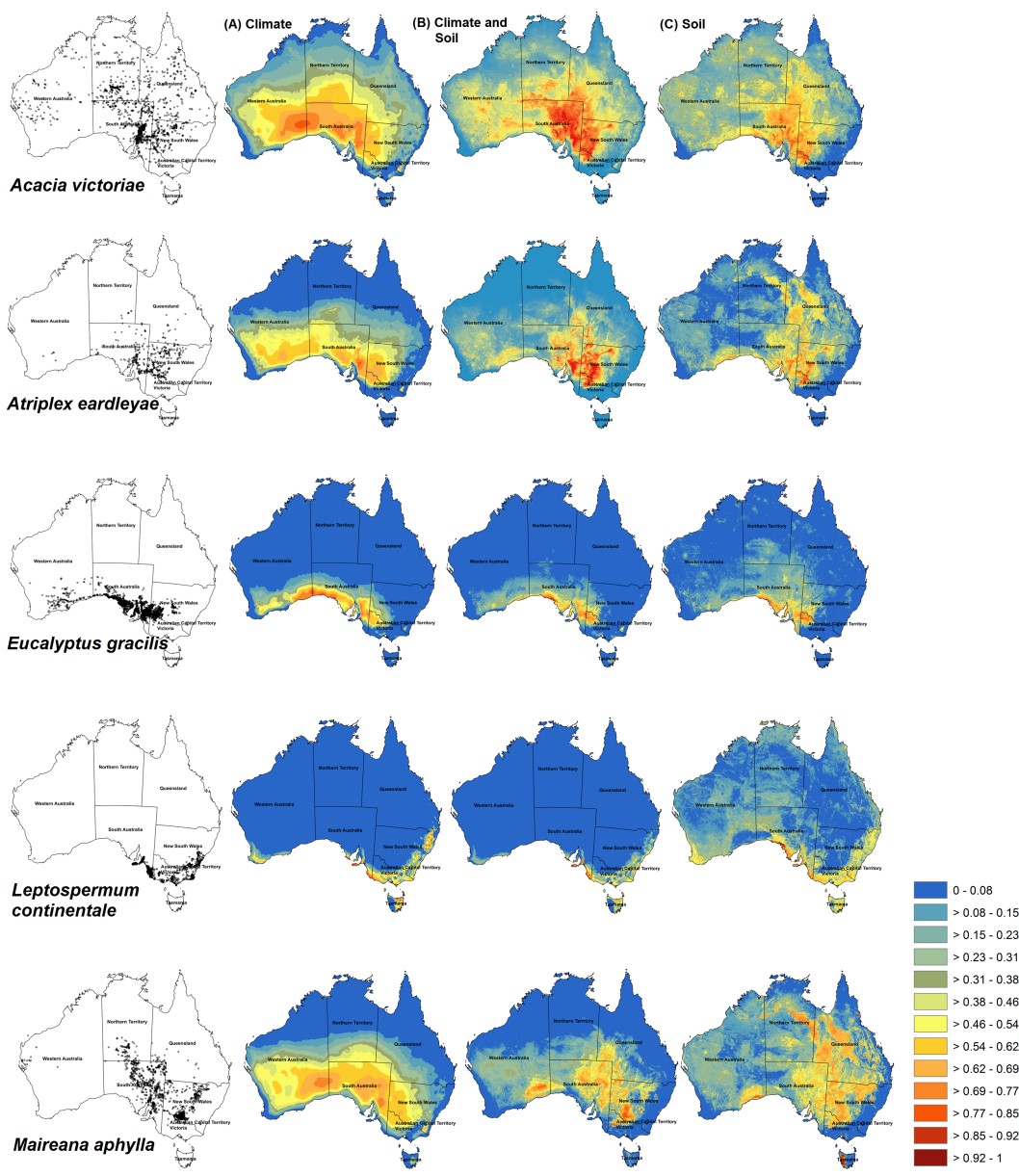

**Figure 2  Habitat prediction maps of some species used in different models.** Maxent predictions of habitat suitability for five Australian shrub species: *Acacia victoriae*, *Atriplex eardleyae*, *Eucalyptus gracilis*, *Leptospermum continentale*, and *Maireana aphylla*. Occurrence records for each species are shown in maps in the first column. Habitat suitability was modelled with different sets of environmental predictors: Climate-only ($V_C$) (column A), Climate-with-soil ($V_{C+S}$) (column B), and Soil-only ($V_S$) (column C). Warmer colours (red) show areas predicted to have higher suitability. Bright blue represents unsuitable areas.

values for TSS were 0.583 (0.100), 0.546 (0.111), and 0.458 (0.010). Among all species, AUC showed a weak relationship with number of environmental variables used in each model, correlation coefficients $|r| < 0.26$ for each of $V_{C+S}$, $V_C$ and $<0.20$ for $V_S$. Visual inspection of maps generated by Maxent indicated that $V_{C+S}$ and $V_C$ models resulted in more realistic projections of habitat suitability than those calibrated with only soil variables (Fig. 2). However, $V_C$ models over-predicted the realized distribution for some

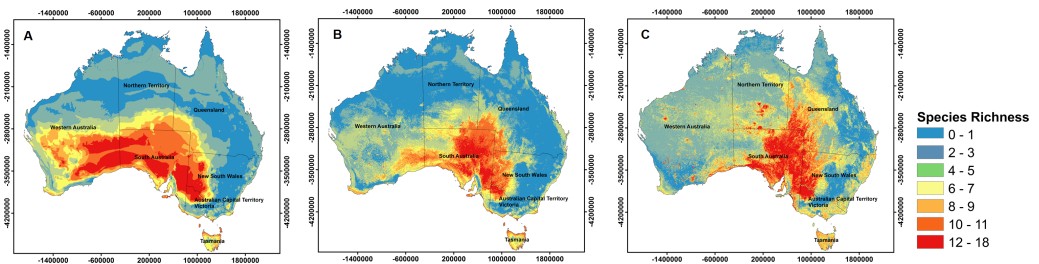

**Figure 3 Species richness based in different used models.** Maps of potential richness of 29 Australian shrub species, based on Maxent models calibrated with (A) climate variables only ($V_C$), (B) climate and soil variables ($V_{C+S}$), and (C) soil variables only ($V_S$). Warmer colours (red) show areas with higher potential richness.

species, whereas $V_{C+S}$ models provided a closer approximation (e.g., *Acacia victoriae* and *Eucalyptus* spp., Fig. 2).

The area of suitable habitat projected by $V_{C+S}$ models ranged from 72,442 km$^2$ (*Leptospermum scoparium*) to 3,736,940 km$^2$ (*Atalaya hemiglauca*). $V_C$ models projected suitable habitat ranging in area from 73,399 km$^2$ (*L. scoparium*) to 5,463,020 km$^2$ (*A. hemiglauca*) (Table 1). Maps from $V_{C+S}$ and $V_C$ models were similar for most species. Exceptions were *A. victoriae, Atriplex eardleyae, Eucalyptus gracilis, Leptospermum continentale*, and *Maireana aphylla* (Fig. 2). In contrast, projections from $V_S$ models tended to cover a smaller spatial extent and had greater fragmentation of suitable habitat.

At a continental scale, the three alternate sets of variables used for model calibration resulted in different patterns of potential richness for the 29 species (Fig. 3). Highest potential richness was associated with $V_S$ models, which predicted a total area of $\sim$135,000 km$^2$ to be suitable for at least 11 (maximum = 18) of the 29 species. However, this map also showed substantial spatial discontinuities (i.e., contiguous areas of high potential richness were smaller than when using other predictor sets). In contrast, $V_{C+S}$ models projected only $\sim$50,000 km$^2$ to be suitable for at least 11 (maximum = 15) species. Although broadly similar patterns were projected by both $V_C$ and $V_{C+S}$ models, the potential richness of shrub species was higher based on $V_C$ models, particularly in central South Australia (Fig. 3).

## Contribution of climate and soil variables to models of shrub distributions

In total, climate variables contributed more to $V_{C+S}$ models than did soil variables for 20 of the 29 species (Table 3). The total contribution of climate variables in $V_{C+S}$ models exceeded 80% for nine species (maximum = 98%, *Leptospermum lanigerum*) and was <20% for one species (18.7%, *E. gracilis*). In contrast, the total contribution of the soil variables in $V_{C+S}$ models exceeded 80% for one species (*E. gracilis*) and was <20% for nine species. Of the individual climate and soil variables, TMwarm and pH were the most influential for ten and eight species, respectively; however, OC contributed the least for 11 species including *Atriplex* and *Eucalyptus* species (Table 3).

For $V_C$ models, TMwarm and T were the most important variables for 14 and 10 species respectively, whereas P contributed the most for five species (Table 3). For models calibrated with soil variables only ($V_S$), pH and BD were the most important variables for

Hageer et al. (2017), *PeerJ*, DOI 10.7717/peerj.3446

**Table 3  Percent contribution of environmental variables used in the different models.** Percent contribution of environmental variables used in the climate-only predictor set ($V_C$), in the climate-with-soil set ($V_{C+S}$), and in the soil-only predictor set ($V_S$) to model 29 Australian shrub species. Mean Annual Temperature ($T$), Maximum Temperature of the Warmest Month (TM warm), Mean Annual Precipitation (P), Precipitation of the Warmest Quarter (PQ warm), Precipitation of the Coldest Quarter (PQ cold), bulk density (BD), clay content percentage (CLAY), pH CaCl$_2$ (pH), and organic carbon (OC). For each species and predictor set, the highest value is shown in bold. For family names of species, see Table 2.

| Species | $V_C$ | | | | | $V_{C+S}$ | | | | | | | | | $V_S$ | | | |
|---|---|---|---|---|---|---|---|---|---|---|---|---|---|---|---|---|---|---|
| | P | T | TM warm | PQ cold | PQ warm | P | T | TM warm | PQ cold | PQ warm | BD | CLAY | pH | OC | BD | CLAY | pH | OC |
| *Ozothamnus turbinatus* | <0.01 | 35.3 | **42.9** | 14.9 | 6.9 | <0.01 | 28.3 | **46.8** | 6.7 | 2.5 | 1.4 | 4.4 | 8.6 | 1.3 | **62.8** | 12.2 | 19.8 | 5.2 |
| *Allocasuarina campestris* | 6.7 | 11.0 | 25.3 | 28.1 | **28.9** | 6.4 | 11.9 | 19.0 | 17.2 | **28.7** | 3.4 | 2.7 | 2.4 | 8.2 | 8.6 | 16.1 | 0.9 | **74.3** |
| *Atriplex angulata* | 45.6 | **45.9** | 5.1 | 0.9 | 2.6 | **37.2** | 33.5 | 6.4 | 0.5 | 1.5 | 7.3 | 6.8 | 6.5 | 0.5 | 24.0 | 20.4 | **37.6** | 18.1 |
| Atriplex eardleyae | 24.3 | 25.1 | 23.5 | 19.8 | 7.3 | 5.9 | 7.5 | 6.5 | 2.0 | 2.3 | 5.0 | 7.2 | 62.9 | 0.7 | 5.9 | 11.2 | **80.5** | 2.4 |
| Atriplex holocarpa | **47.0** | 46.8 | 2.4 | 3.3 | 0.4 | **34.8** | 32.3 | 9.0 | 1.7 | 1.7 | 3.6 | 2.3 | 12.6 | 2.1 | 27.9 | 2.6 | **42.3** | 27.2 |
| Atriplex nummularia | **41.1** | 7.6 | **41.1** | 6.8 | 3.4 | 28.8 | 4.8 | 1.3 | 2.5 | 6.8 | **33.9** | 14.2 | 7.2 | 0.4 | 15.8 | 10.0 | **57.7** | 16.5 |
| Atriplex vesicaria | 36.9 | 1.6 | **49.8** | 3.7 | 7.9 | 29.5 | 2.1 | **32.2** | 1.7 | 9.2 | 9.0 | 2.0 | 14.0 | 0.2 | 8.6 | 7.7 | **77.4** | 6.4 |
| Maireana aphylla | 37.7 | **52.0** | 5.9 | 1.2 | 3.1 | 22.5 | 23.8 | 1.8 | 2.4 | 3.3 | 15.5 | **29.0** | 1.6 | 0.2 | 23.1 | 33.4 | **35.6** | 7.9 |
| *Epacris impressa* | 1.5 | 2.5 | **85.8** | 2.9 | 7.3 | 0.9 | 3.2 | **83.0** | 3.2 | 7.4 | 0.5 | 0.4 | 0.4 | 1.0 | **58.9** | 4.2 | 3.0 | 33.9 |
| Acacia aneura | **30.9** | 15.2 | 29.6 | 21.5 | 2.8 | **27.8** | 10.8 | 4.2 | 7.2 | 1.0 | 26.0 | 4.8 | 10.2 | 8.1 | 18.8 | 18.9 | 26.0 | **36.4** |
| Acacia sclerosperma | 27.7 | **39.3** | 12.1 | 8.2 | 12.7 | 18.4 | **40.0** | 10.1 | 10.3 | 2.6 | 1.5 | 1.5 | 15.4 | 0.3 | 18.6 | 21.1 | **36.6** | 23.7 |
| Acacia tetragonophylla | **70.9** | 6.1 | 2.7 | 3.0 | 17.4 | **26.6** | 6.9 | 4.0 | 0.7 | 10.1 | 13.6 | 4.7 | 22.6 | 10.7 | **34.7** | 30.9 | 10.8 | 23.6 |
| Acacia victoriae | **33.4** | 15.8 | 30.4 | 19.4 | 1.0 | 9.5 | 7.6 | 7.2 | 3.6 | 2.7 | 12.4 | 4.0 | **44.7** | 8.3 | 17.2 | 7.3 | **59.7** | 15.7 |
| Eucalyptus diversifolia | 0.3 | 25.8 | 24.3 | 23.7 | **25.9** | 0.3 | 2.9 | 17.8 | **23.9** | 19.4 | 7.7 | 0.8 | 25.7 | 1.4 | 9.0 | 14.3 | **50.9** | 25.8 |
| Eucalyptus dumosa | 8.2 | **35.4** | 3.4 | 20.2 | 32.9 | 2.2 | 16.1 | 3.5 | 7.9 | 10.2 | 2.6 | 18.0 | **38.6** | 0.9 | 4.4 | 30.7 | **63.9** | 1.0 |
| Eucalyptus gracilis | 15.5 | 20.1 | **37.9** | 7.9 | 18.6 | 1.9 | 4.5 | 4.2 | 4.4 | 3.7 | 0.2 | 8.3 | **72.3** | 0.5 | 0.3 | 12.6 | **86.7** | 0.5 |
| Eucalyptus incrassata | 3.5 | 25.4 | **33.0** | 15.4 | 22.8 | 0.2 | 18.0 | 10.1 | 2.9 | 4.6 | 10.5 | 17.3 | **35.7** | 0.8 | 16.5 | 25.2 | **56.0** | 2.2 |
| Eucalyptus oleosa | 14.2 | 15.3 | **43.4** | 4.2 | 23.0 | 1.7 | 1.7 | 11.3 | 2.8 | 6.3 | 2.5 | 9.2 | **64.3** | 0.2 | 0.1 | 19.1 | **80.5** | 0.3 |
| Eucalyptus socialis | 8.4 | 34.4 | 18.8 | 11.5 | **27.0** | 1.7 | 10.8 | 4.1 | 4.6 | 3.7 | 2.8 | 13.8 | **58.0** | 0.4 | 2.9 | 19.6 | **76.6** | 0.9 |
| Leptospermum continentale | 4.1 | 10.4 | **62.5** | 19.0 | 3.9 | 5.2 | 3.6 | **46.2** | 16.6 | 1.8 | 11.6 | 3.1 | 10.0 | 2.0 | 1.7 | **39.2** | 25.5 | 33.7 |
| Leptospermum glaucescens | 2.9 | **42.4** | 13.4 | 38.7 | 2.6 | 7.3 | **39.6** | 4.4 | 19.7 | 2.1 | 7.2 | 4.4 | 11.1 | 4.2 | 27.7 | 27.2 | **37.4** | 7.6 |
| Leptospermum laevigatum | 9.8 | **31.3** | 28.0 | 26.8 | 4.1 | 1.4 | **26.1** | 23.1 | 20.4 | 1.7 | 0.9 | 12.1 | 8.9 | 5.5 | 14.3 | 19.8 | **38.4** | 27.5 |
| Leptospermum lanigerum | 0.7 | 1.4 | **92.2** | 2.7 | 3.0 | 2.2 | 1.4 | **90.7** | 1.2 | 2.5 | 0.5 | 0.3 | 0.1 | 1.1 | **69.7** | 10.0 | 2.2 | 18.2 |
| Leptospermum scoparium | 0.7 | 12.6 | **83.6** | 1.1 | 2.0 | 0.5 | 12.7 | **80.1** | 1.5 | 2.6 | 0.1 | 1.8 | 0.6 | 0.2 | **35.9** | 30.6 | 17.9 | 15.6 |
| Melaleuca ericifolia | 5.5 | 24.1 | **58.0** | 1.1 | 11.3 | 1.1 | 20.4 | **51.0** | 1.8 | 7.8 | 2.8 | 4.7 | 5.8 | 4.4 | 32.1 | 3.0 | **46.1** | 18.8 |
| Melaleuca squamea | 3.9 | 5.4 | **83.2** | 6.6 | 0.9 | 0.9 | 8.0 | **73.3** | 7.5 | 2.2 | 2.5 | 1.4 | 1.9 | 2.1 | 30.7 | 20.5 | 7.6 | **41.2** |
| Melaleuca squarrosa | 3.1 | 17.6 | **65.9** | 12.3 | 1.0 | 2.7 | 20.9 | **60.7** | 7.5 | 2.2 | 1.7 | 0.3 | 3.3 | 0.8 | **43.1** | 15.3 | 29.1 | 12.4 |
| *Atalaya hemiglauca* | 18.4 | 10.2 | **66.5** | 4.8 | 0.1 | 12.1 | 16.3 | **34.6** | 3.7 | 0.3 | 3.9 | 5.0 | 22.0 | 2.1 | **42.8** | 14.9 | 2.4 | 39.8 |
| *Eremophila freelingii* | 9.0 | **59.6** | 4.9 | 13.0 | 13.5 | 3.0 | **38.1** | 5.0 | 2.4 | 6.2 | 20.3 | 3.7 | 3.4 | 18.0 | **41.2** | 20.2 | 20.5 | 18.1 |

17 and eight species (maximum 86.7% *E. gracilis* and 69.7% *L. lanigerum*), respectively, while OC contributed > 74% to the model for *Allocasuarina campestris* and BD contributed 39.2% to the model for *L. continentale*.

## DISCUSSION

Species distribution models are frequently calibrated only with climate variables, but for plant species, does the addition of soil properties as predictors improve model performance? For 29 Australian shrub species, we found that: (a) on average, models calibrated with both climate and soil variables ($V_{C+S}$) performed better than those calibrated solely with climate variables ($V_C$) (Fig. 1); (b) maximum temperature of the warmest month and pH were the most important contributors to $V_{C+S}$ models for ten and eight species, respectively (Table 3); and (c) models calibrated with only soil variables ($V_S$) had lower AUC and TSS scores, indicating lower classification accuracy than $V_C$ models (Fig. 1), and frequently generated unrealistic predictions (Figs. 2 and 3). For some species the inclusion of soil properties along with climate variables resulted in projections of current habitat that more closely approximated the realized distribution, compared to models calibrated with climate variables only. As a consequence, although broadly similar patterns of potential species richness occurred at the regional level, at finer spatial scales these patterns diverged substantially, particularly in central South Australia (Fig. 3). To date, few studies have explicitly assessed whether the inclusion of soil variables increases the predictive power of SDMs, although a number of studies have included these as variables in model calibration (e.g., *Condit et al., 2013*; *Fitzpatrick et al., 2008*; *Martinson et al., 2011*; *Taylor & Kumar, 2013*; *Zhou et al., 2012*). By themselves, the soil variables included in this study did not result in biologically realistic maps of the realised distribution of the 29 shrub species. The distributions of suitable habitat predicted by these models were frequently fragmented or had abrupt boundaries inconsistent with the known distributions of populations (Fig. 3). Topography and soil type, for instance, are important in determining the suitability of habitat for fire-prone chaparral species in California (*Franklin, McCullough & Gray, 2000*; *Meentemeyer, Moody & Franklin, 2001*). In Australia, *Bui et al. (2014)* found that although climate has higher impact on controlling the distribution of *Acacia* species at a continental scale, physical and chemical properties of soil were more useful in explaining the distribution of shrub species in southern Australia.

At the scale of this study, models indicate that climate plays a greater role than soil characteristics in defining the distribution of most of the 29 shrub species, although soil pH was the key determinant for *Eucalyptus* species. *Bui et al. (2017)* found that incorporating soil variables with climate was efficient for defining the distribution of *Eucalyptus* species and strongly influenced some specific species in taxonomic sections (e.g., Aromatica and Dumaria), although that climate was more important factor. These results are similar to *Martinson et al. (2011)*, who used Maxent to model the distributions of 30 species, including shrubs, across arid areas of North America using climate and soil variables. Temperature variables, mainly T, contributed the most to their models, and none of the three soil variables (pH, clay concentration, and total organic carbon) was the most important for

any species. Variables, such as cation exchange capacity and texture, also contributed little to models predicting the distributions of European trees (*Meier et al., 2012*).

However, soil variables made a substantial contribution to $V_{C+S}$ models for some species. For instance, among *Eucalyptus* species, pH was the highest contributing variable in $V_{C+S}$ models. *Eucalypts* are adapted to live in acidic soils ranging from pH 3.5 to 6 (*Evans, 1992*), through a symbiotic relationship with ectomycorrhizal fungi (*Malajczuk, McComb & Loneragan, 1975*). Conversely, growth of *Acacia* shrubs is restricted in high pH soils due to reductions in nutrient availability (*Nano & Clarke, 2008*). Again, pH contributed highly to $V_{C+S}$ models for several *Acacia* species. Soil bulk density contributed the most to the $V_{C+S}$ model for the saltbush *Atriplex nummularia*, which is known to favour heavy clay soils (*Cunningham & Cunningham, 2011*). Clay soils are found to have lower cation exchange from the organic matter than sandy soil (*McDonald et al., 2017*). This may explain the low contribution of soil organic carbon in the models for *Atriplex* species. These findings highlight the importance of knowledge about specific characteristic and biological idiosyncrasies that species possess to include them as variables predictors.

*González-Orozco et al. (2013)* found annual precipitation and percentage of sand in the topsoil to be key environmental factors influencing the distribution of Australian *Acacia* species. Our results support this, with precipitation contributing the most to $V_{C+S}$ model for *A. aneura* and *A. tetragonophylla*, while bulk density was also important.

## CAVEATS

The accuracy of SDMs is influenced by a number of factors, including accuracy and availability of environmental data used to calibrate models (*Buisson et al., 2010*), biases in occurrence records (*Liu, White & Newell, 2009*), and the selection of model parameters (*Beaumont, Hughes & Pitman, 2008*).

Environmental data frequently require manipulation before use in SDMs, and this often involves resampling data to different resolutions. Aggregation or interpolation to a coarser or finer resolution, respectively, can alter the accuracy of data. In order to match the spatial resolution of the climate data, the soil dataset used in this study was aggregated from 90 m to 5 km. Inconsistencies may have been magnified when the soil data were aggregated to a coarser resolution, and are apparent in some of the maps of suitable habitat. Interpolation and accuracy issues may also arise with climate data. For instance, although new high-resolution climate data (1 km) have recently become available (i.e., eMAST data products; http://www.emast.org.au/), precipitation-related variables may suffer accuracy problems when interpolating to areas with complex topography (*Hutchinson, 1995*). We point out that SDMs calibrated with climate and soil datasets that more closely align in spatial resolution may have different findings to our study. This remains an area requiring further exploration.

It is also likely that patterns in climate and soil do not influence species' distributions at the same spatial scale. For example, different mallee species (*Eucalyptus*) in Western Australia broadly occupy the same hot, dry climatic conditions. Within these climate zones, soil varies at a finer scale. As such, *E. diversifolia* is restricted to the limestone coastal dunes

 

and cliffs, while *E. incrassata* occurs on sand plains such as in South Australia (*Specht, 1966*). Therefore, trade-offs will occur when selecting the most appropriate spatial scale and environmental variables for modelling studies (*Guisan & Zimmermann, 2000*).

An additional hindrance might be the availability of environmental variables that used as predictors in the models. For example, soil temperature and available water capacity of soil that can be absorbed by roots, are suggested to be an important factors that potentially influence plant growth (*Dunne, 1996*; *Reddell, Bowen & Robson, 1985*) and could be incorporated in the SDMs. However, lack of data of these variables at continental scale can be a challenge for using such variables in modelling studies. Furthermore, for some studies that predictions of a variable under alternative scenarios, such as climate change, may be required. Yet these can be difficult to obtain. For instance, while climate change projections for the standard 19 bioclimatic variables included in WorldClim (*Hijmans et al., 2005*) and similar products are readily available, they may not be available for other climate variables.

Accuracy of occurrence records and sampling biases associated with them may affect SDM performance (*Hefley et al., 2013*). Sampling across arid and semi-arid zones of Australia has typically been poor and much clumped in space and time (*Haque et al., in press*). Thus, to reduce the likelihood of errors we applied filters to ALA records to exclude outliers. We decrease sampling bias by removing duplicate records in grid cells and adopted a target-group background approach. Additionally, although we selected dominant, easily-identified species for this study, it is not possible to determine whether their entire realized distribution (and hence, climate envelope) has been sampled.

Regarding species richness, this is a convenient way to describe and compare the biodiversity of different areas; however, there are concerns about over-estimating species richness using combined or so-called stacked SDMs (*Guisan & Rahbek, 2011*; *Hortal et al., 2012*). It is suggested that bias may be corrected by linking stacked SDMs to macroecological models; nevertheless, early comparisons indicate that this approach has not yielded much improvement in reducing overestimates of richness (*Calabrese et al., 2014*). The issue of how best to estimate richness from stacked SDMs will undoubtedly be a key area of research over the next few years.

## CONCLUSIONS

We demonstrate that for most of the shrub species modelled in this study, the inclusion of soil properties, along with climate, resulted in more realistic predictions of the distribution of current habitat. We also demonstrate how maps of suitable habitat can differ substantially depending on whether models are calibrated with only climate variables or with climate and soil variables (e.g., *A. eardleyae* and *E. gracilis*), even when AUC and TSS scores are very similar. This emphasises the importance of interactive validation of model predictions, rather than relying solely on popular metrics for assessing their accuracy. Furthermore, a promising recent application of SDMs is to connect them with stochastic population models to estimate extinction risk (*Keith et al., 2008*; *Stanton et al., 2012*). Such estimates are dependent on reasonable predictions of suitable habitat as a function of climate conditions

and other parameters such as soil type. This is an important consideration when applying SDM methods to scenarios of temporal environmental change (e.g., climate change), whereby projections may continue to diverge. Our analysis validates the approach of incorporating soil variables into SDMs, and we recommend that future studies explore the contribution of soil variables when modelling the distributions of plant species.

**Nomenclature: Australian Plant Census**

| | |
|---|---|
| **AUC** | Area under the receiver operating characteristic curve |
| **BD** | bulk density |
| **CLAY** | percent clay content |
| **P** | total annual precipitation (mm) |
| **OC** | organic carbon |
| **PQ cold** | total precipitation of the coldest quarter (mm) |
| **PQ warm** | total precipitation of the warmest quarter (mm) |
| **SD** | standard deviation |
| **SDM** | species distribution model |
| *T* | mean annual temperature (°C) |
| **TSS** | True Skill Statistic |
| **TM warm** | maximum temperature of the warmest month (°C) |
| $V_C$ | climate-only variable set |
| $V_{C+S}$ | climate-plus-soil variable set |
| $V_S$ | soil-only variable set |

## ACKNOWLEDGEMENTS

We thank Jeremy VanDerWal for assistance with climate data.

### Funding
The authors received no funding for this work.

### Competing Interests
The authors declare there are no competing interests.

### Author Contributions
- Yasmin Hageer conceived and designed the experiments, performed the experiments, analyzed the data, contributed reagents/materials/analysis tools, wrote the paper, prepared figures and/or tables, reviewed drafts of the paper.
- Manuel Esperón-Rodríguez contributed reagents/materials/analysis tools, prepared figures and/or tables, reviewed drafts of the paper.
- John B. Baumgartner analyzed the data, prepared figures and/or tables, reviewed drafts of the paper.
- Linda J. Beaumont conceived and designed the experiments, contributed reagents/materials/analysis tools, reviewed drafts of the paper, interpretaion of results.

## Data Availability

The raw data has been supplied as Data S1.

## Supplemental Information

Supplemental information for this article can be found online at http://dx.doi.org/10.7717/peerj.3446#supplemental-information.

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
