# Peer review of "Climate, soil or both? Which variables are better predictors of the distributions of Australian shrub species?"

_PeerJ, doi:10.7717/peerj.3446_

## Round 0.1 · original submission · Major Revisions

The paper was evaluated in detail by 3 reviewers, and all of them agree that this is a nice paper. Reviewer 1 and 2 noted that the paper is well organised and suggested improvements. Reviewer 1 also questioned the use of limited and correlated soil variables (bulk density and carbon content). In addition, the soil map was created by regression equations relating climate, topography and vegetation information.

In the reviews, some additional references were suggested, but of course the authors should only include them if they agree that they strengthen the paper.

I hope the authors can address all of these concerns.

Reviewer 1 ·

Basic reporting

This paper addresses an important question, namely the relative role of climate and edaphic factors on shrubby plant species distribution in Australia. The paper is written clearly. Unfortunately the authors have not done a thorough investigation of the existing literature on this subject and have overlooked numerous key papers (see list below for a few).

Key Australian literature:
Beadle, N. C. W. (1966). Soil phosphate and its role in molding segments of the Australian flora and vegetation, with special reference to xeromorphy and sclerophylly. Ecology, 47(6), 992-1007.
Bui, E. N. (2013). Soil salinity: A neglected factor in plant ecology and biogeography. Journal of Arid Environments, 92, 14-25.
Bui, E. N., & Henderson, B. L. (2013). C: N: P stoichiometry in Australian soils with respect to vegetation and environmental factors. Plant and Soil, 373(1-2), 553-568.
Bui, E. N., González-Orozco, C. E., & Miller, J. T. (2014). Acacia, climate, and geochemistry in Australia. Plant and Soil, 381(1-2), 161-175.
Cowling, R. M., & Witkowski, E. T. F. (1994). Convergence and non‐convergence of plant traits in climatically and edaphically matched sites in Mediterranean Australia and South Africa. Austral Ecology, 19(2), 220-232.
Cowling, R. M., Witkowski, E. T. F., Milewski, A. V., & Newbey, K. R. (1994). Taxonomic, edaphic and biological aspects of narrow plant endemism on matched sites in mediterranean South Africa and Australia. Journal of Biogeography, 651-664.
Fonseca, C. R., Overton, J. M., Collins, B., & Westoby, M. (2000). Shifts in trait‐combinations along rainfall and phosphorus gradients. Journal of Ecology, 88(6), 964-977.
González‐Orozco, C. E., Laffan, S. W., Knerr, N., & Miller, J. T. (2013). A biogeographical regionalization of Australian Acacia species. Journal of Biogeography, 40(11), 2156-2166.
González-Orozco, C. E., Laffan, S. W., & Miller, J. T. (2011). Spatial distribution of species richness and endemism of the genus Acacia in Australia. Australian Journal of Botany, 59(7), 601-609.
Hodgkinson, K. C., & Oxley, R. E. (1990). Influence of fire and edaphic factors on germination of the arid zone shrubs Acacia aneura, Cassia nemophila and Dodonaea viscosa. Australian Journal of Botany, 38(3), 269-279.
Nano, C. E., & Clarke, P. J. (2008). Variegated desert vegetation: Covariation of edaphic and fire variables provides a framework for understanding mulga‐spinifex coexistence. Austral Ecology, 33(7), 848-862.
Sander, J., & Wardell‐Johnson, G. (2011). Fine‐scale patterns of species and phylogenetic turnover in a global biodiversity hotspot: Implications for climate change vulnerability. Journal of Vegetation Science, 22(5), 766-780.
Tongway, D. J., & Ludwig, J. A. (1990). Vegetation and soil patterning in semi‐arid mulga lands of eastern Australia. Austral Ecology, 15(1), 23-34.

Experimental design

Because the literature is not well consulted, the research question is then poorly implemented because the authors do not understand how to use the information available and key soil properties such as % sand, nitrogen, phosphorus, and available water content are not used. This is unfortunate because they are available from the Soil Landscape Grids of Australia (SLGA). Of the four SLGA data layers used, why is the 0-5 cm depth interval the only one used when there are six depths available (l. 167)—even using the logic ‘that most of the nutrients are concentrated in the top ~20 cm’, they should have used the top three layers and calculated a depth-weighted average for 0-20cm for each of the four edaphic properties. pH is a measure of acidity, not salinity (l. 165)—there is no salinity layer in SLGA. BD and SOC are highly correlated, so why use both?
Otherwise the statistical methods, using Maxent to model, AUC, and TSS to evaluate SDM results, are ok. How do the ‘potential high species richness’ areas compare to the NVIS shrublands?

Validity of the findings

The findings are valid given the data used. However the soil data is sub-optimal.

The discussion would also improve if the literature listed above (and other references therein) is consulted.

·

Basic reporting

I have now reviewed the manuscript, “Climate, soil or both? Which variables are better predictors of the distributions of Australia shrub species”. Overall, I think this is an interesting study (given shrub species are generally a forgotten and barely studied taxonomic group) and a well-written manuscript. The use of proximal environmental predictors (i.e. physiologically-relevant variables) and in particular, the use of variables accounting for soil properties is rare in SDM literature and I believe studies showing how their use may improve the accuracy of predictions are very much needed (especially studies advocating for looking beyond the use of bioclim variables as the only predictors of species distributions).

The introduction is clear but too long. Despite the main aim of the study is to demonstrate that the use of soil variables in conjunction with climate variables may increase the predictive performance of SDMs, there is only one paragraph in the introduction talking about the scarce use of predictors other than climate SDM literature (lines 67-79) and the reasons why this is the case. The authors focus way too much on talking about the taxonomic group aim of study (shrubs). I think the introduction would benefit from (1) trimming down/summarizing the part of the description of the ecology of the shrub species (maybe the description of the Australian species/genus considered in the study could be moved to the beginning of the Methods section) (2) adding a paragraph in which authors introduce why and how the use of variables other than climate (and especially soil properties in the case of modelling plants) may improve the predictive performance of models.
I disagree with the authors in that topographic and land use variables are scarcely used in SDMs literature. In studies other than global of continental scales, topography is used to downscale climatic data; this is, models using climate variables derived from Regional Climate Models (RCMs) do - at the very least- include topography indirectly (elevation and aspect). Land use variables are scarcely used in impact assessment studies making future predictions of biodiversity (Titeaux et al. 2016) but widely used in current predictions of species distributions (although most of the time people refer to land use as land cover, leading to confusion between the two concepts).

The figures and tables are relevant and well described. I only wonder why the authors did not show the variable contributions for the models Vs (based on soil variables) in the main manuscript (Table 3) so the readers can fully compare the three model sets without need of consulting the supplementary material (Vc, Vs and Vc+s). On the other hand, variable contributions for the Vc models is presented both in Table 3 and Supplementary material S1 (remove duplication?).

Raw data and R code are supplied so results could initially be reproduced.

Ref.
Titeux N., Henle K., Mihoub J.-B., Regos A., Geijzendorffer I.R., Cramer W., Verburg P.H. & Brotons L. (2016) Biodiversity scenarios neglect future land-use changes. Global Change Biology, 22, 2505-2515.

Experimental design

The data is robust and the methods are generally sound but I believe some aspects have to be clarified before the manuscript could be considered for publication.

(1) I am not concerned about the use of AUC as a measure of predictive performance in terms of models´ discrimination ability in presence-only SDMs. However, AUC can potentially be influenced by the number of covariates and possible over-fitting. The authors report that the average number of records per species is 3,214 but this average value could be achieved with a mix of species with very large datasets ( >10,000 records) and some species with a few records (<40 records). The authors report the links to the raw data in the supplementary material (which is fantastic) but some clarification about the exact number of final records available to fit the models after data cleaning might be helpful to assess if the models for some species could be potentially overfitted. I suggest adding an additional column to the ‘raw data’ table indicating this number.
Also, the authors report that the Vc+s models showed the highest predictive performance across most modelled species. However, this higher AUC values might possibly only be reflecting the fact that Vc+s models were fit using almost double number of predictors (9) than the the Vc and Vs models (5). This is neither mentioned in the methods section or discussed in the ‘caveats’ section. I wonder whether the authors would find the same results if all model type sets (Vc, Vs and Vc+s) were based on the same number of candidate predictors. This shouldn´t be hard to test given the authors used a very high threshold to exclude highly correlated variables (0.85 vs 0.7 recommended in other studies; e.g. Dormann et al. 2013): some climatic variables with Pearson´s correlation coefficients > 0.7 and < 0.85 could be excluded from the list of candidate predictors without losing much information.

(2) The strength of this study is the availability of soil data at a relatively high resolution ~90 x 90 m. However, for the modelling exercise soil data is resampled to 5 x 5 km to match the spatial resolution of the climate data, losing the fine scale detail that the authors were originally defending soil data could provide to models. The caveats of this decision are discussed by the authors but I believe there is no need of losing so much detail on soil data to fit the models. Despite interpolation of rainfall data might be problematic, authors could use relatively simple approaches to interpolate/refine temperature values to higher resolutions (e.g. 1 km or finer) using an adiabatic lapse rate and a digital elevation model of the desired resolution (Kearny et al. 2014 and Morán-Ordóñez et al. 2017 used this method to refine the same AWAP data the authors use in this study). If models could be fit at 1 km or finer resolution (even assuming rainfall patterns do not change within a 5 km cell) inconsistencies linked to resampling of soil data would be smaller and models would be more realistic of the realised distribution of the shrub species (and potentially - this is my guess- soil predictors would contribute more to explain model variance). One of the references recommended here is my own. I would like to clarify I’m not looking for a citation here; I just pointed this article out because we used a different lapse rate than Kearny et al. (2014) to refine temperature values from AWAP data.

(3) Thresholding predictions losses information and there are studies that demonstrate it is best o use continuous predictions of habitat suitability instead (Guillera-Arroita et al., 2015). I wonder if observed results of pair-wise differences in predicted area between model sets would be the same if choosing a threshold different than the ‘maximum training sensitivity plus specificity’. The authors could sum the logistic predictions of each species across the landscape (corresponding biogeographic regions) to compare the suitable habitat predicted by the different model types instead of using a single threshold or at least, they should present a sensitivity analysis of results to the chosen threshold.

Ref.

Dormann, C. F., Elith, J., Bacher, S., Buchmann, C., Carl, G., Carré, G., ... & Münkemüller, T. (2013). Collinearity: a review of methods to deal with it and a simulation study evaluating their performance. Ecography, 36(1), 27-46

Guillera‐Arroita, G., Lahoz‐Monfort, J. J., Elith, J., Gordon, A., Kujala, H., Lentini, P. E., ... & Wintle, B. A. (2015). Is my species distribution model fit for purpose? Matching data and models to applications. Global Ecology and Biogeography, 24(3), 276-292.

Kearney, M. R., Shamakhy, A., Tingley, R., Karoly, D. J., Hoffmann, A. A., Briggs, P. R., & Porter, W. P. (2014). Microclimate modelling at macro scales: a test of a general microclimate model integrated with gridded continental-scale soil and weather data. Methods in Ecology and Evolution, 5(3), 273-286.

Morán-Ordóñez, A., Briscoe, N. J., & Wintle, B. A. (2017). Modelling species responses to extreme weather provides new insights into constraints on range and likely climate change impacts for Australian mammals. Ecography. DOI: 10.1111/ecog.02850.

Validity of the findings

The discussion of the manuscript is well linked to the research question but it repeats in large extent what it is shown in the results section (lines 267-278). Also, the only species group for which there is some discussion about why soil properties (pH) were important model predictors is the Eucalypts genus. As a reader, I am also interested in knowing the authors' hypotheses about why the soil variables were relatively unimportant for other species (apart from the plausible inconsistencies due to resampling soil data –see my previous comment). Do you think there are soil properties that are more important for the modelled species but for which there is no available data? which ones? could the importance of the soil variables relate to a particular species’ trait that differs across the species groups modelled here? Are the results just reflecting the scale of the modelling exercise (continental vs regional or local?)?

·

Basic reporting

This paper reports on a species distribution modeling effort to understand, between climate variables and soil variables that are individually and in combination are the best predictors of the distribution of 29 selected Australian shrub species. The paper as presented conforms with the stated Peerj format and guidelines.

The paper is well written. Good sentence structure is used throughout and there are no obvious grammar or spelling issues.

Experimental design

The research question and hypothesis is clearly stated and is relevant given previous research and current prevailing knowledge in this area. The methodology is a very standard approach for this problem. The authors use the common model MAXENT for this project. Input data sets are also in line with current approaches. The spatial resolution of the data used for the project is appropriate and also in line with current practice.

Validity of the findings

The conclusions of the paper are mostly in line with results from similar efforts. The authors determined that adding soil properties as predictors along with climate variables does incrementally improve model performance. Climate has been shown to be the dominant factor in previous work and this was shown to be the case in this work as well. Climate and climate + soil variables generated similar patterns, while soil-only predictor variables produced results of a more fragmented nature.

The author’s caveats (beginning with Line 305) do point out the challenge of re-sampling higher-resolution soils information for an exercise such as this. The authors do also point out, very importantly, that climate and soil properties do not vary at the same spatial scale.
Overall, the conclusions of the authors, based upon the modeling effort undertaken, are valid.

An “elephant” in the room may be topography. I’m curious as to why this fairly significant variable, that has been applied in other vegetation SDM’s, was not mentioned or included in the analysis? In fact, I’m wondering if topography when added with climate and soils could be an important intermediary predictor variable—effectively bridging the length scale differences of climate and soil. I would be interested to see some acknowledgment of topography in the occurrence of shrub species and the reasoning behind lack of discussion or inclusion in the study. Perhaps there are very compelling reasons and, if so, a mention would prove valuable. Certainly, topographic information, at the appropriate scale, would be available for inclusion in a study of this type.

The conclusions section did sum up the finding of the work, but I felt it could be been enhanced with a bit more discussion about where to head next with this type of SDM effort. Are there other modifications to the approach that could improve the results? Which specific variables should be included in new studies? Are there variables (like topography) that MAY be valuable if tested?

---

## Round 0.2 · Minor Revisions

The paper was evaluated again by one of the original reviewers, which recommended a minor revision. The authors have not fully revised the manuscript as indicated.

The author also need to address in their paper the use of limited and correlated soil variables (bulk density and carbon content). In addition, the soil map was created by regression equations relating climate, topography and vegetation information.

In the reviews, some additional references were suggested, but of course the authors should only include them if they agree that they strengthen the paper.

I hope the authors can address all of these concerns.

Reviewer 1 ·

Basic reporting

The authors need to incorporate some the responses to my comments regarding the SLGA data layers selected into the main body of the paper.

They claim to have changed error in "pH is a measure of acidity" (l. 165) but have not.

There are 2 newly published papers that they should use in their discussion and acknowledge:
Bui et al. (2017) Climate and geochemistry as drivers of eucalypt diversification in Australia. Geobiology. DOI: 10.1111/gbi.12235
Prentice, E., Knerr, N., Schmidt-Lebuhn, A.N. et al. (2017) Do soil and climate properties drive biogeography of the Australian Proteaceae? Plant and Soil. doi:10.1007/s11104-017-3261-6

Experimental design

ok

Validity of the findings

ok

Annotated reviews are not available for download in order to protect the identity of reviewers who chose to remain anonymous.

---

## Round 0.3 · accepted · Accept

Thank you for addressing the reviewer's comments, although it is clearly shown that BD is strongly correlated with OC (r = -0.79), which is the point made by the reviewer, you should just use one factor (OC).